# The Effectiveness of Smoking Cessation, Alcohol Reduction, Diet and Physical Activity Interventions in Improving Maternal and Infant Health Outcomes: A Systematic Review of Meta-Analyses

**DOI:** 10.3390/nu13031036

**Published:** 2021-03-23

**Authors:** Louise Hayes, Catherine McParlin, Liane B Azevedo, Dan Jones, James Newham, Joan Olajide, Louise McCleman, Nicola Heslehurst

**Affiliations:** 1Population Health Sciences Institute, Newcastle University, Newcastle upon Tyne, NE2 4AX UK & Fuse, The Centre for Translational, Research in Public Health, North East England, UK; nicola.heslehurst@ncl.ac.uk; 2Department of Midwifery, Nursing and Health, Northumbria University, Newcastle upon Tyne, NE1 8ST UK & Fuse, The Centre for Translational, Research in Public Health, North East England, UK; catherine.mcparlin@northumbria.ac.uk; 3School of Human and Health Sciences, University of Huddersfield, Huddersfield HD1 3DH, UK; l.azevedo@hud.ac.uk; 4School of Health and Life Sciences, University of Teesside, Middlesbrough TS1 3BX, UK; dan.jones@tees.ac.uk (D.J.); joan.olajide@nhs.net (J.O.); 5Faculty of Health and Life Sciences, Northumbria University, Newcastle upon Tyne, NE1 8ST UK & Fuse, The Centre for Translational Research in Public Health, North East England, UK; james.newham@northumbria.ac.uk; 6Department of Midwifery, Nursing and Health, Northumbria University, UK & Fuse, The Centre for Translational Research in Public Health, North East England, UK; 7Population Health Sciences Institute, Newcastle University, Newcastle upon Tyne NE2 4AX, UK; louise-800@hotmail.com

**Keywords:** systematic review, intervention, pregnancy, smoking, alcohol, diet, physical activity

## Abstract

Diet, physical activity, smoking and alcohol behaviour-change interventions delivered in pregnancy aim to prevent adverse pregnancy outcomes. This review reports a synthesis of evidence from meta-analyses on the effectiveness of interventions at reducing risk of adverse health outcomes. Sixty-five systematic reviews (63 diet and physical activity; 2 smoking) reporting 602 meta-analyses, published since 2011, were identified; no data were identified for alcohol interventions. A wide range of outcomes were reported, including gestational weight gain, hypertensive disorders, gestational diabetes (GDM) and fetal growth. There was consistent evidence from diet and physical activity interventions for a significantly reduced mean gestational weight gain (ranging from −0.21 kg (95% confidence interval −0.34, −0.08) to −5.77 kg (95% CI −9.34, −2.21). There was evidence from larger diet and physical activity meta-analyses for a significant reduction in postnatal weight retention, caesarean delivery, preeclampsia, hypertension, GDM and preterm delivery, and for smoking interventions to significantly increase birth weight. There was no statistically significant evidence of interventions having an effect on low or high birthweight, neonatal intensive care unit admission, Apgar score or mortality outcomes. Priority areas for future research to capitalise on pregnancy as an opportunity to improve the lifelong wellbeing of women and their children are highlighted.

## 1. Introduction

There are multiple risks for maternal, fetal and infant health associated with maternal behaviours during pregnancy. The prevention of adverse pregnancy outcomes is critical to the immediate health of the mother and infant, as well as having an impact on lifelong health. Several adverse pregnancy outcomes, for example maternal and perinatal mortality, low birthweight and preterm birth, share common risk factors (smoking, alcohol consumption, and diet and physical inactivity, particularly in the context of obesity) [1,2,3,4,5,6,7,8]. There are also specific health risks to certain maternal behaviours, such as alcohol consumption and Fetal Alcohol Spectrum Disorders [9,10,11], and diet and physical activity behaviours and the development of gestational diabetes mellitus (GDM), particularly among women living with obesity [12,13,14,15]. Interventions to modify maternal behaviours, including reducing smoking and alcohol consumption, and improving diet and physical activity, in order to prevent adverse pregnancy outcomes are widespread. National and international guidelines exist for weight management and smoking cessation [16,17,18,19]. However, guidance on alcohol consumption is variable, with advice ranging from abstinence to light consumption [20,21,22].

Despite the plethora of behaviour-change interventions in pregnancy, there is a lack of data relating to whether there are similarities or differences between interventions targeting specific behaviours in the prevention of adverse health outcomes among mothers and their infants. Interventions in pregnancy, and subsequent synthesis of evidence regarding interventions, tend to be carried out in silos without an attempt to synthesise across behaviours, despite the overlap in target population, health professionals responsible for delivering multiple interventions, and comparable health outcomes. Synthesising evidence across behaviours enables the exploration of any shared evidence-base on the effectiveness of interventions in pregnancy. The shared mechanisms of what types of intervention are effective and for whom, and consistencies and inconsistencies in the existing evidence-base, are needed for the identification of future research and practice directives.

This paper is the second to be reported from a wider programme of systematic reviews of systematic reviews exploring diet, physical activity, smoking and alcohol behavioural interventions delivered during pregnancy. The wider aims of this programme of systematic reviews of systematic reviews are to (1) examine the effectiveness of interventions on changing maternal behaviours in pregnancy, (2) examine the effectiveness of interventions on improving health-related outcomes for women and infants, and (3) explore any shared behavioural techniques or content of interventions that may be associated with effectiveness across these behaviours. We have published a systematic review of systematic reviews addressing aim 1 [23]. This paper reports a systematic review of systematic reviews addressing aim 2. 

## 2. Materials and Methods

The methods used for this systematic review of systematic reviews are published in the first paper in this programme of systematic reviews [23]. The methods are also summarised here, alongside details on the search updates and amendments to the inclusion criteria specific to the context of the aim of this paper. The purpose of carrying out a systematic review of systematic reviews is to provide an overview of existing systematic reviews, to compare the findings to identify research gaps and provide a direction for future research. We used the Joanna Briggs Institute (JBI) methodology for umbrella reviews [24] and the PRISMA reporting guidelines and checklist (Appendix A) [25]. The protocol for this systematic review has been published [26], and was registered on the PROSPERO database (CRD42016046302).

### 2.1. Identification of Studies

A comprehensive search of fourteen bibliographic databases was originally conducted in May 2016, and updated in March 2018, November 2019 and September 2020 (Appendix A). The search was limited to English language reviews published over the previous 10 years to yield primary research conducted 30+ years prior to the reviews [24]. An additional search of Google Scholar was carried out in November 2020 (Appendix A) to identify any additional systematic reviews not identified by the bibliographic database searches. EndNote reference management software was used to manage the results and screening. Following de-duplication, titles and abstracts were screened against the inclusion and exclusion criteria. Full-text screening used a pre-defined template (Appendix A) and reasons for exclusion were recorded. All stages of screening were carried out by two reviewers independently (including NH, JN, LM, LA, CM, JO, LH, DJ) with any disagreements discussed and a third reviewer available for arbitration if required. 

### 2.2. Inclusion and Exclusion Criteria

The inclusion criteria are based on population, intervention, comparator, outcome, and study design (PICOS). The population (P) was pregnant women. We excluded systematic reviews reporting the effectiveness of interventions delivered during the preconception or postnatal periods, or as treatments for existing conditions (e.g., as a treatment for women diagnosed with GDM). The interventions (I) needed to target maternal smoking, alcohol, diet or physical activity behaviours. Pharmacological and dietary supplement interventions were excluded. Systematic reviews were not excluded based on the type of comparator groups used by the included interventions (e.g., where comparator groups may have been no intervention, or a different type of intervention) (C). The outcomes (O) of interest were the effectiveness of interventions at preventing adverse health-related outcomes for the mother and infant. Systematic reviews were excluded if they only reported the effectiveness of interventions on maternal target behaviours during pregnancy without reporting the effectiveness on any health-related outcomes. The study design (S) included systematic reviews reporting a meta-analysis of at least two studies. 

### 2.3. Data Extraction and Quality Assessment

All systematic reviews which met the inclusion criteria involved data extracted using a standardised template developed for this research. The standardised JBI critical appraisal instrument for umbrella reviews [24] was used to assess the methodological quality of included reviews (Appendix A for full details). The data extraction and quality assessment tools were piloted by a group of reviewers (NH, JN, LH, JO, LM, LA, DJ) and refined to improve consistency between reviewers. All data extraction and quality assessments were carried out by one reviewer (non-blind) and validated by a second reviewer (including LH, CM, LM, LA, DJ, NH, JO). Any discrepancies between the original reviewer and the validation reviewer were resolved by discussion.

### 2.4. Evidence Synthesis

The evidence synthesis prioritises searching for consistency in the reported effectiveness of interventions at improving health-related outcomes for women and infants across the reviews and behaviours, and the identification of gaps in the existing evidence-base. The meta-analysis results reported by the included reviews are presented in tabular format according to the reported outcomes and type of intervention. Results data include: the pooled estimate (e.g., risk ratio (RR), odds ratio (OR) or mean difference (MD)); corresponding measures of variance (e.g., 95% confidence interval (CI)); statistical significance; and direction of effect (for significant results only). Where possible, forest plots were created to visually summarise the meta-analysis results. Each line on the plot is the pooled result from a meta-analysis reported by the included systematic reviews, grouped according to the type of intervention. 

All tables and forest plots are accompanied by a narrative overview of the systematic review characteristics and findings [24] using a systematic narrative synthesis approach [27]. First, we identified maternal and infant outcome categories; for example, a category was created for “maternal weight-related outcomes” which included total and weekly gestational weight gain (GWG), excessive, adequate and inadequate GWG, postnatal weight retention, and any other outcome measures relating to maternal weight during pregnancy or postnatally. Tables for outcome categories were collated and results within the tables were stratified according to the behaviour (e.g., smoking, diet), type of intervention (e.g., incentives, low glycaemic index (GI) diet), and any population sub-groups (e.g., based on maternal body mass index (BMI) being in the recommended, overweight or obese ranges, usually defined as 18.5–24.9 kg/m^2^, 25.0–29.9 kg/m^2^ and ≥30.0 kg/m^2^ respectively). An overview of the significance of results, direction of effect and range in effect sizes is provided. Similarities and differences between the significant and non-significant results were explored in the context of the behaviours and types of interventions or population sub-groups. 

## 3. Results

### 3.1. Included Systematic Reviews

A total of 26,898 unique records were identified by database searches and 500 results were screened from the Google Scholar searches. Of these, 121 were systematic reviews of relevant behaviour-change interventions delivered in pregnancy and published since 2011; 22 smoking, 3 alcohol, 95 diet and/or physical activity and one that included both smoking and diet and/or physical activity interventions (Figure 1). Fifty-six were excluded as they did not report meta-analysis of health-related outcomes, leaving 65 which met the inclusion criteria and were included in this review (Appendix A). There were 63 [28,29,30,31,32,33,34,35,36,37,38,39,40,41,42,43,44,45,46,47,48,49,50,51,52,53,54,55,56,57,58,59,60,61,62,63,64,65,66,67,68,69,70,71,72,73,74,75,76,77,78,79,80,81,82,83,84,85,86,87,88,89,90] reviews which reported various combinations of diet and physical activity interventions (66% of the 95 diet and physical activity systematic reviews we identified in the search) and two [91,92] for smoking-cessation interventions (9% of the 22 smoking reviews identified), whereas none of the three alcohol systematic reviews reported a meta-analysis of health-related outcomes.

Out of the 63 included reviews that reported interventions with different diet- and/or physical-activity-related components, 19 reported effectiveness data for diet-only interventions [30,35,39,40,42,44,45,51,52,55,62,63,65,67,69,82,85,89,90], 37 reported data for physical-activity-only interventions [33,36,38,39,42,46,47,48,49,51,52,53,54,55,59,62,66,68,69,70,71,72,73,74,75,76,78,80,81,82,83,84,85,87,88,89,90], and 37 reported effectiveness data for combined diet and/or physical activity interventions [28,29,30,31,32,34,37,39,40,41,43,45,50,51,52,55,56,57,58,60,61,62,64,69,70,71,72,73,74,77,79,81,82,85,86,89,90]. Some reviews reported data for more than one category; therefore, these categories totalled more than 63. The diet and/or physical activity group included reviews which had pooled data in the meta-analysis for diet and/or physical activity components, for diet-only interventions, and for physical-activity-only interventions.

The majority of included systematic reviews reported searching databases plus supplementary searches for all types of intervention (*n* = 2 smoking, 34 diet and/or physical activity, 18 diet only and 35 physical activity only); supplementary searches primarily involved searching trial registers and databases, followed by searching reference lists of included studies or related systematic reviews, and, to a lesser extent, hand searching and contacting authors (Table 1 and Appendix A).

The pooled sample sizes in the included reviews ranged from 214 to 598,185 women (Table 1 and Appendix A); there was a median pooled sample size of 6920 women in systematic reviews reporting on a meta-analysis of diet and/or physical activity interventions, 8558 women for diet-only interventions and 4350 women for physical-activity-only interventions. Most systematic reviews were restricted to included studies with an RCT design (*n* = 100% for smoking, 65% for diet and/or physical activity, 95% for diet only, and 73% for physical activity only; see Appendix A for details on alternative study designs included). The number of intervention studies included in the systematic reviews ranged from three to 113, with a similar median number of studies across all combinations of diet and physical activity (*n* = 21 to 23).

The studies included in the reviews were published between 1974 and 2019. The countries included in the intervention studies within the systematic reviews were reported for both smoking systematic reviews. However, there were data missing from six of the systematic reviews reporting diet and/or physical activity outcomes, six for diet only and seven for physical activity only. All systematic reviews that reported the countries of intervention setting for their included studies had pooled the data from multiple countries, and none reported meta-analysis data for a single setting (Appendix A). When reported, according to the World Bank classification [93], studies were predominantly set in High-Income Countries and Upper Middle-Income Countries; 25 High-Income Countries and 13 Upper-Middle-Income Countries were represented across all included reviews (Table 1 and Appendix A). Lower-Middle-Income Countries were not represented in any smoking systematic reviews, whereas the different combinations of diet and physical activity interventions were based in four Lower-Middle-Income Countries. One Low-Income Country was included for diet-only interventions, whereas none of the other types of intervention included any Low-Income Countries.

#### 3.1.1. Quality

The included systematic reviews were predominantly high-quality (88%), with no systematic reviews rated as low-quality (Table 2, Appendix A). Out of a maximum possible score of 11, scores ranged from six (moderate quality) to 11 (high quality). A maximum score of 11 was achieved by 17 of the included systematic reviews, whereas only one review scored six (Appendix A). When comparing the systematic reviews that reported results for different types of intervention, the maximum quality score was achieved by 12 of the 37 reviews reporting results for diet and/or physical activity, eight of the 19 reporting diet only, and five of the 37 reviews reporting results for physical activity only. The maximum score achieved by smoking systematic reviews was 10. All systematic reviews across all behaviours had clearly and explicitly stated questions and used appropriate methods to combine studies (100% for questions 1 and 8), and a high percentage (>70%) scored positively for the remaining questions, except for question 6 (was critical appraisal conducted by two or more reviewers independently) which scored 69% (Table 2). As there were only two included smoking reviews, there should be caution in the interpretation of the percentages for this type of intervention. When looking at the different intervention types for diet- and physical-activity-related interventions, all scored highly for all questions, with the exception of physical-activity-only interventions, where only 59% had two researchers carry out critical appraisal independently (Table 2, question 6).

#### 3.1.2. Overlap of Included Studies

A systematic review of systematic reviews will include duplications of original studies reported by multiple reviews. There were 120 citations of included intervention studies in the two smoking reviews (Appendix A); after the removal of duplicate citations of the same publications across multiple reviews, there were 116 unique publications remaining (Appendix A). The 63 diet and physical activity reviews had a total of 1871 citations for included intervention studies across all systematic reviews (Appendix A), of which 675 were citations of unique publications (Appendix A).

### 3.2. Maternal Health Outcomes

There were 332 meta-analyses of maternal health outcomes reported by 57 included systematic reviews; these were related to maternal weight, GDM, hypertensive disorders, mode of delivery, and “other” maternal outcomes (Table 3). The majority of the data reported were for maternal weight-related outcomes (*n* = 38 systematic reviews, 114 meta-analyses), these were primarily total GWG (*n* = 66 meta-analyses), followed by GDM (*n* = 32 systematic reviews, 73 meta-analyses). Meta-analyses were most frequently reported for diet and/or physical activity interventions (*n* = 174), followed by physical-activity-only interventions (*n* = 97). There were no systematic reviews reporting meta-analysis data on the effectiveness of smoking interventions at preventing or improving any maternal health outcomes.

#### 3.2.1. Maternal Weight-Related Outcomes

There were 38 systematic reviews, reporting 114 meta-analyses of weight-related outcomes [28,29,31,32,33,34,37,38,40,41,43,47,48,49,50,51,52,53,55,56,60,61,63,64,65,68,69,75,77,78,79,81,82,83,85,86,87,89] (Appendix A), including total and weekly GWG, excessive, adequate and inadequate GWG (using Institute of Medicine criteria, Appendix A), postnatal weight retention, and “other” measures of maternal weight. 

There were 66 meta-analyses of mean GWG, reported by 36 systematic reviews [28,29,31,32,33,34,37,38,41,43,47,48,49,50,51,52,53,55,56,60,61,63,64,65,68,69,75,78,79,81,82,83,85,86,87,89], with a general pattern for a reduction in GWG among women who received intervention compared with controls (Figure 2, Appendix A). The majority of meta-analyses (*n* = 53) showed significantly reduced GWG (Appendix A). There was little difference between significant and non-significant results in relation to the direction of effect, with only one study reporting an increased GWG for women in the intervention arm (MD 0.28 kg (95% CI −1.13, 1.69) [29]). There were 32 meta-analyses which combined diet and/or physical activity interventions, with 26 reporting significantly reduced mean GWG (ranging from −0.21 kg (95% CI −0.34, −0.08) [79] to −4.65 kg (95% CI −8.14, −0.56) [28]) and six reporting a non-significant change in mean GWG (ranging from −1.62 kg (95% CI −3.57, 0.33) [61]) to 0.28 kg (95% CI −1.13, 1.69) [29]). There were 10 meta-analyses which were limited to diet-only interventions, and eight reported a significantly reduced mean GWG (ranging from −1.56 kg (95% CI −2.94, −0.99) [82] to −5.77 kg (95% CI −9.34, −2.21) [89]) and two non-significant results (−0.69 kg (95% CI −1.74, 0.36) [65]) and −0.72 kg (95% CI −1.48, 0.04) [55]). Twenty-four meta-analyses were restricted to physical-activity-only interventions and 19 showed a significantly reduced mean GWG (ranging from −0.36 kg (95% CI −0.64, −0.09) [49] to −2.22 kg (95% CI −3.13, −1.30) [33]) while five were non-significant (ranging from −0.58 kg (95% CI −1.30, 0.13) [81]) to −0.12 kg (95% CI −0.51, 0.26) [49]). 

Overall, 38 meta-analyses included women with any BMI (or combined recommended, overweight and obese BMI, excluding underweight), and 32 showed a significantly reduced mean GWG (ranging from −0.21 kg (95% CI −0.34, −0.08) [79] to −4.70 kg (95% CI −8.07, −1.34) [63]), while six showed non-significant effects (ranging from −1.62 kg (95% CI −3.57, 0.33) [61] to −0.28 kg (95% CI −0.64, 0.09) [31]). Among the meta-analyses that reported specific BMI groups, three out of five reported a significantly reduced mean GWG for women with a recommended BMI (MD ranging from −0.77 kg (95% CI −1.15, −0.39) [55] to −1.61 kg (95% CI −1.99, −1.22) [78]); one out of two for overweight BMI (MD −0.75 kg (95% CI −1.22, −0.27) [55]); 13 out of 16 for overweight or obese BMI (MD ranging from −0.36 kg (95% CI −0.64, −0.09) [49] to −5.77 kg (95% CI −9.34, −2.21) [89]); and all three for obese BMI reported significantly reduced GWG (ranging from −0.85 kg (95% CI −1.41 to −0.29) [55] to −4.65 kg (95% CI −8.14, −0.56) [28]). 

The effectiveness of diet and/or physical activity interventions on weekly GWG was reported by four systematic reviews [29,50,60,81], with a general pattern for a reduction in GWG among women in the intervention arm (Appendix A) (Figure 3). Three meta-analyses showed significantly reduced mean kg/week weight gain for women in intervention arms compared with controls (ranging from −0.03 kg/week (95% CI −0.06, −0.00) [60] to −0.26 kg/week (95% CI −0.42, −0.09) [50]), while one non-significant result showed a similar effect size (−0.04 kg/week (95% CI −0.11, 0.04) [29]. No data for weekly GWG were reported for diet- or physical-activity-only interventions, or for individual BMI categories. 

Meta-analysis data for excessive GWG according to the Institute of Medicine recommendations was reported by nine systematic reviews [29,40,41,50,51,52,60,61,81] and 12 meta-analyses (Appendix A); inadequate GWG was reported by five systematic reviews [29,40,41,60,81] and seven meta-analyses (Appendix A); and adequate GWG was reported by three systematic reviews [41,60,81] and five meta-analysis (Appendix A). All categories of weight gain had data for diet and/or physical activity and physical-activity-only interventions, whereas only excessive GWG had data for diet-only interventions. There was a pattern across the meta-analyses for a reduction in excessive GWG, and an increase in both adequate and inadequate GWG among women who received the intervention compared with controls (Figure 4A–C). There was a significant reduction in excessive GWG in half of the meta-analyses (ranging from OR 0.68 (95% CI 0.59, 0.78) [81] to RR 0.87 (95% CI 0.79, 0.96) [60]) and non-significant results had similar effect sizes (ranging from OR 0.76 (95% CI 0.13, 4.59) [61]) to RR 0.90 (95% CI 0.77, 1.05) [50]) (Figure 4A). The increase in adequate GWG was significant in three meta-analyses (ranging from OR 1.39 (95% CI 1.16, 1.67) to OR 1.69 (95% CI 1.19, 2.42) [81]) but not significant for four meta-analyses (ranging from RR 1.00 (95% CI 0.86, 1.18) [29] to 1.33 (95% CI 0.74, 2.37) [41]) (Figure 4B). The increase in inadequate GWG below the Institute of Medicine recommendations was significant in three meta-analyses (ranging from RR 1.14 (95% CI 1.02, 1.27) [40] to OR 1.32 (95% CI 1.04, 1.67) [81]) but not significant for two meta-analyses (RR 1.02 (95% CI 0.93, 1.11) [60]) and RR 1.32 (95% CI 0.96, 1.83) [41]) (Figure 4C). There were limited data for BMI subgroups across all categories of GWG.

Sixteen meta-analyses for postnatal weight retention were reported by seven systematic reviews for diet and/or physical activity interventions ([28,29,40,50,60,77,81], and one for physical-activity-only [81] (Appendix A). No data were available for diet-only interventions. There was a general pattern for a reduction in postnatal weight retention for women who received the intervention, with the exception of one meta-analysis, which showed a significantly increased weight retention at 6 weeks postnatal (weighted (W) MD 0.58 kg (95% CI 0.13, 1.03) [50]), and one which was non-significant (MD 1.05 kg (95% CI −2.73, 4.83) [29]) (Figure 5, Appendix A). Nine meta-analyses showed a significant reduction in postnatal weight retention at different follow-up time periods. The mean difference in postnatal weight retention ranged from −0.68 kg (95% CI −1.28, −0.09) at 12 months [77] to −1.90 kg (95% CI −1.69, −1.12) at 6 months [50], and the risk of postnatal weight retention as a binary outcome was also significantly reduced (RR 0.78 (95% CI 0.63, 0.97) [40]). Non-significant reductions in mean postnatal weight retention were reported in five meta-analyses (ranging from −0.38 kg (95% CI −1.12, 0.35) [77] to −1.12 kg (95% CI −2.49, 0.25) [40]). Limited data were available for BMI sub-groups, with one meta-analysis showing significantly reduced postnatal weight retention among women with a recommended BMI [29], while two showed no significant difference for women with overweight or obesity [29,77]. 

Additional weight-related outcomes were reported in three systematic reviews [28,51,60] for diet and/or physical activity interventions (Appendix A). These were BMI at delivery, postnatal weight loss, postnatal BMI and postnatal return to pre-pregnancy BMI. Only postnatal return to pre-pregnancy BMI was significantly increased for women who had received diet and/or physical activity interventions during pregnancy (relative risk (RR) 1.25 (95% CI 1.08, 1.45) [60]). No additional weight-related outcomes were reported for diet-only or physical-activity-only interventions, or for BMI subgroups.

#### 3.2.2. Gestational Diabetes Related Outcomes

There were 32 systematic reviews reporting 73 meta-analyses outcomes related to GDM [29,35,36,39,41,42,43,45,46,47,50,51,52,53,54,55,56,58,60,61,62,63,65,66,68,72,74,75,78,80,88,90] (Appendix A). A GDM diagnosis was reported by 29 systematic reviews [29,35,36,39,41,42,43,45,46,47,50,51,52,53,54,55,56,58,60,62,63,66,68,72,75,78,80,88,90] and 59 meta-analyses (*n* = 26 diet and/or physical activity, *n* = 14 diet-only, *n* = 19 physical-activity-only, Appendix A). The direction of effect was generally towards reduced odds of GDM in women who received the intervention compared with controls (*n* = 54) (Figure 6), although there was inconsistency in statistical significance. A significantly reduced risk of developing GDM for women who received interventions compared with controls was found by 22 meta-analyses: five for diet and/or physical activity, six for diet-only interventions, and 11 physical-activity-only interventions (ranging from OR 0.33 (95% CI 0.14, 0.76) [39] to RR 0.83 (95% CI 0.69, 1.00) [62]) (Figure 6). Non-significant effects were reported for 37 meta-analyses: 21 for diet and/or physical activity, and eight for both diet-only and physical-activity-only interventions. Thirty-two of the non-significant meta-analyses reported a reduced effect similar to the significant results (ranging from RR 0.52 (95% CI 0.27, 1.03) [51] to RR 0.99 (95% CI 0.83, 1.19) [56]), while five showed an increased effect (ranging from RR 1.02 (95% CI 0.41, 2.57) [41] to OR 1.44 (95% CI 0.96, 2.14) [39]).

Among the meta-analyses that reported specific BMI groups, two out of seven results for recommended BMI showed a reduced risk of GDM (RR 0.51 (95% CI 0.31, 0.82) [54] and RR 0.58 (95% CI 0.37, 0.90) [78]; both were physical-activity-only interventions); four out of 14 for overweight or obese BMI (ranging from RR 0.40 (95% CI 0.18, 0.86) [45] to RR 0.83 (95% CI 0.69, 1.00) [62]); one obese-only meta-analysis was not significant [60]. 

There were an additional 14 meta-analyses reported by seven systematic reviews for other outcomes related to GDM [39,42,61,65,66,74,80] (Appendix A). The majority of outcomes were for blood glucose (*n* = 10), of which four reported significant reductions among women who received low-glycemic index (GI) diets, diet and/or physical activity, or physical-activity-only interventions (ranging from WMD −0.18 mmol/L (95% CI −0.33, −0.02) [65] to MD −0.48 mmol/L (95% CI −0.76, −0.19) [74]), while six reported non-significant results (ranging from SMD 0.01 (95% CI −0.34, 0.36) [80] to MD −1.02 mmol/L (95% CI −2.75, 0.71) [66]). There were limited data available for the other outcomes, and there were no significant results for HbA1C [61,65], fasting plasma insulin [80] or insulin use [65]. There were limited data available for specific BMI sub-groups. Only one review [80] reported data for overweight or obese BMI and physical-activity-only interventions and found no significant difference between intervention and control arms for fasting plasma glucose or insulin. 

#### 3.2.3. Hypertensive Disorders

There were 22 systematic reviews [29,30,35,39,40,41,45,50,51,52,53,54,55,56,57,60,63,66,68,72,75,82] reporting 59 meta-analyses for outcomes related to hypertensive disorders (Appendix A), including preeclampsia, pregnancy-induced hypertension, and “other” outcomes. Intervention effectiveness at preventing preeclampsia was reported by 17 systematic reviews [29,30,39,40,41,45,50,51,52,53,60,63,66,68,72,75,82] and 27 meta-analyses (*n* = 15 diet and/or physical activity, *n* = 6 diet only, *n* = 6 physical activity only, Appendix A). There were also 15 systematic reviews [35,39,40,41,45,51,52,54,55,60,63,68,72,75,82] that reported 27 meta-analyses for pregnancy-induced hypertension (*n* = 12 diet and/or physical activity, *n* = 8 diet only, *n* = 7 physical activity only, Appendix A). Although the general direction of effect was towards a reduced risk of preeclampsia among women who participated in interventions compared with controls (20 out of 27 meta-analyses), only five showed a significantly reduced risk; two for diet and/or physical activity interventions (both reporting a RR of 0.74 (95% CI 0.59, 0.92 and 0.60, 0.92 [51,52]), two for diet-only (both reporting a RR of 0.67 (95% CI 0.53, 0.85) [30,51,52]) and one for physical-activity-only (OR 0.59 (95% CI 0.37, 0.94) [72]) (Figure 7A). The 22 non-significant results ranged from an RR of 0.34 (95% CI 0.10, 1.22) [41,60]) to 1.39 (95% CI 0.66, 2.93) [75]. Similar to preeclampsia, there was an overall direction of effect towards a reduction in hypertensive disorders of pregnancy (26 out of 27 meta-analyses), although only ten were statistically significant; four for diet and/or physical activity intervention, three for diet-only and three for physical-activity-only (ranging from RR 0.21 (95% CI 0.09, 0.45) [54] to OR 0.85 (95% CI 0.71, 1.00) [55]), Figure 7B). The 17 non-significant results ranged from a RR 0.46 (95% CI 0.16, 1.29) [60] to an OR of 0.95 (95% CI 0.78, 1.16) [55]. Nine meta-analyses were restricted to specific BMI subgroups for preeclampsia, and none were significant. However, for pregnancy-induced hypertension, two meta-analyses for the recommended BMI subgroup showed significantly reduced risk [41,54], while six for overweight or obese were not significant. 

There were four systematic reviews [35,56,57,60] that reported five additional meta-analyses relating to “other” measures of hypertensive disorders (Appendix A). There were three meta-analyses of diet and/or physical activity interventions and composite outcomes for preeclampsia and pregnancy-induced hypertension (one of which showed a significant reduction in risk [57], whereas the other was non-significant [56]) and severe preeclampsia, HELLP syndrome and eclampsia, which was non-significant [60]. One systematic review [35] also reported significant reductions in both systolic and diastolic blood pressure for diet-only interventions (SMD −0.26 mmHg (95% CI −0.45, −0.07) and SMD −0.57 mmHg (95% CI −0.75, −0.38), respectively).

#### 3.2.4. Mode of Delivery Outcomes

There were 25 systematic reviews [29,35,36,39,40,41,43,45,50,51,52,54,55,56,57,58,60,63,65,68,73,75,78,83,84,87] reporting 63 meta-analyses of mode of delivery-related outcomes, including caesarean delivery, the induction of labour, instrumental vaginal delivery, and vaginal delivery (Appendix A). There were 42 meta-analyses of caesarean delivery reported by 25 systematic reviews [29,35,36,39,40,41,43,45,50,51,52,54,55,56,57,58,60,63,65,68,73,75,78,83,84,87]. There was an overall pattern towards a reduced direction of effect (38 out of 42 meta-analyses), of which only seven showed a significantly reduced risk (*n* = 4 diet and/or physical activity interventions, *n* = 3 physical-activity-only intervention, *n* = 0 diet-only interventions) with ORs ranging from 0.80 (95% CI 0.69, 0.94) [87] to 0.91 (95% CI 0.83, 0.99) [55]; Appendix A). Non-significant results ranged from RR 0.78 (95% CI 0.58, 1.05) [84] to RR 1.33 (95% CI 0.97, 1.84) [36]. Eleven meta-analyses restricted to specific BMI categories and results were similar to those which included women of any BMI (Figure 8). 

There were eight meta-analyses of induction of labour and diet and/or physical activity or diet-only interventions, reported by seven systematic reviews [29,40,45,51,52,60,63,83] (Appendix A), and the overall direction of effect was towards an increased risk among women receiving the interventions (7 out of 8), although only one showed a significantly increased risk (RR 1.12 (95% CI 1.00, 1.26) [51,52]) (Figure 9A). The non-significant results ranged from an RR of 0.92 (95% CI 0.79, 1.06) [60] to 1.14 (95% CI 0.54, 2.40) [45]. There were six meta-analyses of instrumental vaginal delivery and diet and/or physical activity or physical-activity-only interventions, reported by five systematic reviews [36,54,60,73,84] (Appendix A). There was an overall reduction in odds (five out of six meta-analyses), although only one statistically significant (OR 0.76 (95% CI 0.63, 0.92) [73]) (Figure 9B). Non-significant results ranged from RR 0.78 (95% CI 0.61, 1.01) [54] to RR 1.07 (95% CI 0.86, 1.34) [60]. There were seven meta-analyses of vaginal delivery reported by four systematic reviews [51,52,54,84] (Appendix Ad), with no consistent pattern in the overall direction of effect. Two physical-activity-only meta-analysis found significantly increased odds of vaginal delivery among women receiving the intervention (OR 1.09 (95% CI 1.04, 1.15) [54] and RR 1.12 (95% CI 1.01, 1.24) [84]) (Figure 9C), whereas five meta-analyses encompassing all categories of diet and/or physical activity intervention found no significant effect (ranging from RR 0.97 (95% CI 0.89, 1.07) [51,52] to RR 1.02 (95% CI 0.93, 1.11) [52]). No data were available for overweight or obese BMI subgroups for the induction of labour, instrumental delivery or vaginal delivery outcomes. 

#### 3.2.5. Other Maternal Health Outcomes

There were eight systematic reviews [40,51,52,55,56,60,63,71] that reported 23 additional meta-analyses relating to maternal health (Appendix A). These were postpartum haemorrhage (PPH, *n* = 4), low back pain (*n* = 1), perineal trauma (*n* = 1), prenatal mental health measures (*n* = 9), postnatal mental health measures (*n* = 5) and a composite measure of adverse maternal health outcomes (*n* = 3). Four out of nine meta-analyses of prenatal mental health measures reported a significant reduction in depression (OR 0.33 (95% CI 0.21, 0.53) and 0.55 (95% CI 0.34, 0.90) [71]) and depressive symptoms (SMD −0.23 (95% CI −0.36, −0.09) and −0.38 (95% CI −0.51, −0.25) [71]) among women of any BMI receiving diet and/or physical activity or physical-activity-only interventions, whereas one meta-analysis found no significant effect on depression in women with an overweight or obese BMI (MD in score −0.06 (95% CI −0.29, 0.17) [56] and four found no significant association with state anxiety, or state or trait anxiety symptoms in women with any BMI [71]. There were no other significant effects of interventions for any of the other maternal health outcomes reported. 

### 3.3. Infant Health Outcomes

There were 270 meta-analyses of infant health outcomes reported by 36 included systematic reviews; these were related to fetal growth, gestational age at delivery, mortality, admission to the neonatal intensive care unit (NICU), Apgar score, and “other” infant health-related outcomes (Table 4). The majority of the data reported was for fetal-growth-related outcomes (*n* = 33 systematic reviews, 150 meta-analyses), followed by outcomes related to gestational age at delivery (*n* = 26 systematic reviews, 55 meta-analyses). All categories of outcomes were reported by the three combinations of diet and physical activity interventions, whereas meta-analyses of smoking interventions were only available for fetal growth, gestational age, mortality and NICU outcomes. Meta-analyses were most frequently reported for diet and/or physical activity interventions (*n* = 117), followed by similar numbers of physical-activity-only interventions (*n* = 69) and diet-only interventions (*n* = 61), while meta-analyses of smoking interventions were the least reported (*n* = 23). 

#### 3.3.1. Fetal Growth Outcomes

There were 33 systematic reviews [28,29,36,39,40,41,42,43,44,45,50,51,52,53,54,55,56,57,58,59,60,63,65,66,67,68,75,76,78,79,87,91,92] that reported 150 meta-analyses of fetal-growth-related outcomes (Appendix A) including birth weight, large for gestational age (LGA), macrosomia >4000 g or >4500 g, small for gestational age (SGA), low birthweight (LBW) <2500 g, and “other” measures of fetal growth. 

The effectiveness of interventions on increasing or decreasing birth weight was reported by 30 systematic reviews [28,29,36,39,40,41,43,44,45,50,51,52,53,54,56,57,58,59,60,63,65,66,67,68,75,76,78,87,91,92] and 45 meta-analyses (*n* = 6 smoking, *n* = 14 diet and/or physical activity, *n* = 9 diet only, *n* = 16 physical activity only, Appendix A). All meta-analyses of smoking cessation interventions had a direction of effect towards interventions to increase birthweight, and the majority (four out of six) showed a statistically significant increase (ranging from MD 0.28 g (95% CI 0.05, 0.50) [92] to 134.58 g (95% CI 76.32, 192.83) [91]) (Figure 10). The two non-significant meta-analyses showed a MD of 56.02 g (95% CI −31.46, 143.50) and 79.43 g (95% CI −53.05, 211.91) [91]. However, there was no consistent direction of effect, and limited significant findings among meta-analyses of the different categories of diet and physical activity interventions. Six meta-analyses showed a significantly reduced birthweight (ranging from WMD −31 g (95% CI −57 to −4) [87] to MD −0.84 g (95% CI −1.16, −0.52) [40], one significantly increased (SMD 0.19 (95% CI 0.06, 0.31) [67]), and 32 meta-analyses were not statistically significant (ranging from MD −151.08 g (95% CI −528.90, 226.73) [76] to MD 28.24 g (95% CI −78.26, 134.74) [29]. Thirteen meta-analyses were restricted to specific BMI groups, and none were significant for any intervention type. 

In relation to the prevention of high-birthweight outcomes, there were 16 systematic reviews [29,39,40,42,43,51,52,53,55,60,65,67,68,75,76,79] that reported 29 meta-analyses for intervention effect on LGA (*n* = 14 diet and/or physical activity, *n* = 8 diet only, *n* = 8 physical activity only, Appendix A), 13 systematic reviews [29,36,39,40,41,42,50,56,58,60,65,67,75] that reported 16 meta-analyses for intervention effect on macrosomia (*n* = 9 diet and/or physical activity, *n* = 4 diet only, *n* = 3 physical activity only, Appendix A), and one systematic review [87] that reported two meta-analyses for a composite large-at-birth outcome and physical-activity-only interventions (Appendix A). There were no data available for smoking interventions. For all high-birthweight outcomes, the direction of effect was towards a reduction in high birthweight for women in intervention arms compared with control (42 out of 47 meta-analyses, Appendix A), although statistically significant findings were not observed in most reviews. Five out of 29 meta-analyses found a significant reduction in LGA among women who received interventions compared with controls; two for diet and/or physical activity interventions, two for diet-only (both meta-analyses of low-GI interventions), and one for physical-activity-only (ranging from RR 0.14 (95% CI 0.05, 0.41) [42] to RR 0.73 (95% CI 0.54, 0.99) [51]) (Figure 11A). The 24 non-significant results for LGA ranged from RR 0.37 (95% CI 0.06, 2.30) [51] to 1.25 (95% CI 0.50, 3.11) [40]. Only two meta-analyses found a significant reduction in macrosomia for women participating in interventions compared with controls; one for physical-activity-only interventions and macrosomia defined as birthweight >4000 g (RD 0.36 (95% CI 0.13, 0.99) [42]), and one for diet and/or physical activity interventions and birthweight >4500 g (RR 0.63 (95% CI 0.42, 0.94) [60]) (Figure 11B). The 14 non-significant results for macrosomia ranged from RR 0.89 (95% CI 0.78, 1.01) [60] to 2.19 (95% CI 0.63, 7.60) [41]. One meta-analysis of the large at birth composite outcome showed a significantly reduced odds for any BMI (OR 0.69 (95% CI 0.55, 0.86) [87]), but this was not significant for overweight or obese BMI (OR 0.71 (95% CI 0.36, 1.41) [87]). A further six LGA and four macrosomia meta-analyses were restricted to BMI-specific groups, and none of these were significant. 

The prevention of low birthweight (LBW) was reported by nine systematic reviews [29,40,41,50,54,58,67,91,92] and 13 meta-analyses (*n* = 6 smoking, *n* = 5 diet and/or physical activity, *n* = 1 diet-only, *n* = 1 physical-activity-only, Appendix A) with inconsistent direction of effect. One smoking and one diet-only meta-analysis showed significantly reduced LBW (OR 0.65 (95% CI 0.42, 0.88) [92] and SMD −0.19 (95% CI −0.32, −0.05) [67], respectively), and there were no significant results for diet and/or physical activity or for physical-activity-only interventions (Figure 12A). The 11 non-significant meta-analyses ranged from RR 0.58 (95% CI 0.32, 1.04) [91] for smoking cessation interventions, to 1.30 (95% CI 0.8, 2.10) [50] for diet and/or physical activity interventions. A further 11 systematic reviews [29,39,40,51,52,53,55,60,65,67,75] reported 21 meta-analyses for SGA (*n* = 8 diet and/or physical activity, *n* = 7 diet-only, *n* = 6 physical-activity-only, Appendix A) (Figure 12B), and one systematic review [87] reported two meta-analyses for a composite small-at-birth outcome and physical-activity-only interventions (Appendix A) with no significant results for any type of intervention (ranging from OR 0.90 (95% CI 0.31, 2.63) [87] to RR 1.49 (95% CI 0.47, 4.71) [40]. There were limited data for BMI subgroups, with one meta-analysis for SGA, three for LBW, and one for small-at-birth; none of which were significant.

There were seven systematic reviews [51,60,63,65,67,76,87] that reported 22 meta-analyses of the effectiveness of diet and/or physical activity (*n* = 10), diet-only (*n* = 10) and physical-activity-only (*n* = 2) interventions on additional measures of fetal/neonatal growth (Appendix A). Outcomes reported were ponderal index (*n* = 3), fetal/infant fat mass (*n* = 4), skinfold thickness (*n* = 1), abdominal circumference (*n* = 3), birth length (*n* = 7) and head circumference (*n* = 4). Only three meta-analyses were significant, all of them were diet-only interventions. One review found significantly reduced fetal fat mass (MD −0.04 kg (95% CI −0.06, −0.01) [51]) among the infants of women who had participated in dietary interventions during pregnancy compared with controls. There were conflicting results for infant length, with one review identifying a significant reduction (MD −1.84 (95% CI −3.61, −0.08) [51]), while another identified a significant increase (SMD 0.08 cm (95% CI 0.01, 0.15) [67]). There were no meta-analyses for additional measures of fetal growth, which were restricted to specific BMI subgroups or smoking interventions.

#### 3.3.2. Gestational Age at Delivery Outcomes

There were 26 systematic reviews reporting 55 meta-analyses outcomes related to gestational age at delivery [28,29,35,36,39,40,45,50,51,52,53,54,55,56,57,58,59,60,63,65,66,68,75,78,87,91] (Appendix A). Outcomes were mean gestational age in weeks or days and preterm delivery. Twenty-three meta-analyses of mean gestational age at delivery were reported by 18 systematic reviews [28,29,35,36,50,51,52,53,54,56,57,59,60,63,65,66,78,87] (Appendix A). There were eight meta-analyses for diet and/or physical activity interventions, five meta-analyses of diet-only intervention, and 10 for physical-activity-only, with no consistent direction of effect (Figure 13). Two meta-analyses showed a significantly increased mean gestational age at delivery (WMD 0.22 weeks (95% CI 0.01, 0.42) [50] and MD 0.05 week (95% CI 0.07, 0.17) [54]). The 21 non-significant meta-analyses ranged from MD −0.07 weeks (95% CI −0.29, 0.16) [53] to 0.20 weeks (95% CI −0.02, 0.42) [52]. There were limited data for specific BMI subgroups; one meta-analysis for recommended BMI showed a significant increase in mean gestational age [54], and two for overweight or obese BMI were not significant [56,87]. No data were reported for smoking interventions. 

There were 32 meta-analyses reported by 18 systematic reviews [29,35,39,40,45,50,51,52,54,55,58,60,63,65,66,68,75,91] for preterm delivery (Appendix A). There was one systematic review for smoking interventions [91], which reported five meta-analyses of the effectiveness of different types of intervention content (counselling, feedback and incentives) and, although there was a pattern of a reduction, none showed any significant effect on preterm delivery (Figure 14). Amongst the categories of diet and physical activity interventions, there was a different pattern in direction of effect for physical-activity-only interventions (increased) compared with diet and/or physical activity or diet-only interventions (reduced), although limited studies found statistically significant results throughout. There were 11 meta-analyses for diet and/or physical activity interventions and eight diet-only, of which six showed a significantly reduced risk of preterm delivery among women who received the intervention compared with controls (ranging from OR 0.28 (95% CI 0.08, 0.96) [55] to RR 0.80 (95% CI 0.65, 0.98) [60]). The 13 non-significant results ranged from RR 0.33 (95% CI 0.11, 1.02) [40] to OR 1.20 (95% CI 0.45, 3.15) [39]. None of the eight meta-analyses for the physical-activity-only interventions were significant (ranging from OR 0.93 (95% CI 0.44, 1.99) [66] to 1.29 (95% CI 0.90, 1.85) [55]). There were four meta-analyses which were restricted to specific BMI subgroups, and only one was statistically significant for a reduced risk of preterm delivery among women with an overweight or obese BMI (RR 0.62 (95% CI 0.41, 0.95) [58]). 

#### 3.3.3. Mortality Outcomes

There were 17 meta-analyses reported by 10 systematic reviews [51,52,55,58,60,63,67,68,70,91] for mortality outcomes (Appendix A) including stillbirth (*n* = 7 smoking, diet and/or physical activity and diet-only interventions), intrauterine death (*n* = 2 diet and/or physical activity and physical-activity-only interventions), neonatal mortality (*n* = 2 smoking and diet and/or physical activity interventions), and perinatal mortality (*n* = 4 diet and/or physical activity, diet-only, physical-activity-only interventions). There was no consistent direction of effect or significant effect of any type of intervention on any of the mortality outcomes reported in the meta-analyses.

#### 3.3.4. Neonatal Intensive Care Unit Admission

There were 14 meta-analyses reported by seven systematic reviews [29,45,51,52,55,60,91] for admission to NICU (Appendix A). One smoking systematic review reported three meta-analyses for different types of interventions and control groups, there were seven meta-analyses for diet and/or physical activity interventions, one for diet-only, and three for physical-activity-only interventions, including one which was limited to women with an overweight or obese BMI. None of the results showed any significant differences between intervention and control arms for NICU admission (Figure 15A).

#### 3.3.5. Apgar Score

There were seven systematic reviews [36,51,57,59,60,66,67] that reported 11 meta-analyses related to Apgar score (Appendix A). Outcomes were Apgar score <7 at 5 minutes (*n* = 5), Apgar score at 1 minute (*n* = 2) and Apgar score at 5 minutes (*n* = 4). There were no significant effects of interventions reported for any Apgar score outcome for diet and/or physical activity, diet-only or physical-activity-only interventions (Figure 15B). There were no data available for BMI subgroups or for smoking interventions.

#### 3.3.6. Other Infant Health-Related Outcomes

There were eight systematic reviews [40,45,51,52,55,60,67,73] that reported 23 additional meta-analyses relating to other infant-health-related outcomes (Appendix A). These were shoulder dystocia (*n* = 5), neonatal/infant hypoglycaemia (*n* = 4), respiratory distress syndrome (*n* = 3), infant hyperbilirubinaemia (*n* = 2), birth trauma (*n* = 2), placental weight (*n* = 1), premature rupture of membranes (PROM, *n* = 1), breastfeeding at 6 months (exclusive *n* = 1, partial *n* = 1), and a composite measure of adverse offspring outcomes (*n* = 3). Two out of five meta-analyses reported a significantly reduced risk of shoulder dystocia among women who participated in diet and/or physical activity (RR 0.39 (95% CI 0.22, 0.70) [51,52]) or diet-only interventions (RR 0.38 (95% CI 0.21, 0.69) [51,52]), and the non-significant results ranged from RR 1.02 (95% CI 0.57, 1.83) [40] to RR 1.24 (95% CI 0.81, 1.91) [45]. Two out of three meta-analyses for respiratory distress syndrome reported a significantly reduced risk among women who participated in diet and/or physical activity interventions (RR 0.47 (95% CI 0.26, 0.85) [40] and RR 0.56 (95% CI 0.33, 0.97) [60]), whereas the non-significant result was RR 1.05 (95% CI 0.48, 2.28) [51,52]. There were no other significant effects of interventions for any of the other infant-health-related outcomes reported. There was no data reported for smoking interventions, or for subgroups of BMI and other infant-health-related outcomes.

### 3.4. Conflict of Interest

Given the nature of the topics of the reviews and potential conflicts of interest, particularly relating to industry funding, we have summarised whether any conflicts of interest were reported by review authors (Appendix A). Out of the 65 included systematic reviews, only one [75] did not include any conflict-of-interest statement in the published paper (reporting meta-analysis of physical-activity-only interventions Appendix A). Of the 64 reviews that made a declaration of potential conflicts of interest, 59 reported that there were no conflicts of interest, or that they had received funding from organisations where no industry-related conflict of interest was deemed to be present (e.g., from non-profit organisations, research councils or national public health/health service agencies). Authors of five systematic reviews reported potential conflicts of interest, with one reporting smoking-cessation interventions [91] (Appendix A), three reporting diet and/or physical activity interventions [29,34,60] (Appendix A) and one reporting diet-only interventions [65] (Appendix A). Four reported being authors on related reviews or included studies within the systematic review [29,34,60,91]. Three reported receiving funding from related pharmaceutical, diagnostic or food industries for activities that were unrelated to the systematic reviews [34,60,91]. One reported that the authors were employees of a research laboratory funded by the food industry [65].

## 4. Discussion

This systematic review of systematic reviews provides a critical overview of the existing evidence-base on the effectiveness of behaviour-change interventions in pregnancy on improving health-related outcomes for women and infants. We identified a high volume of high-quality published evidence, with 65 included systematic reviews reporting 602 meta-analyses, of which 57 reviews reported 332 meta-analyses of maternal health-related outcomes and 36 reported 270 meta-analyses of infant-health-related outcomes. The most frequently reported maternal-health-related outcomes were related to maternal weight followed by GDM, and for the infant outcomes most frequently related to fetal growth and gestational age at delivery. The evidence synthesis identified the strongest and most consistent evidence-base for interventions to significantly reduce total and weekly GWG. There was also conflicting evidence reported for some outcomes relating to statistical significance, where the larger meta-analyses (greater number of included studies and pooled number of participants) appear to suggest a positive effect of interventions on some maternal and infant health outcomes that was not consistent across all results. For example, the statistically significant meta-analyses for excessive GWG included more studies (mean 11 vs. 4) and had more participants pooled in the analysis (mean 2144 vs. 581) than non-significant meta-analyses. Similar patterns were observed for other maternal outcomes (including adequate and inadequate GWG, postnatal weight retention, GDM, preeclampsia, hypertensive disorders of pregnancy and caesarean delivery), and for birthweight in the infant outcomes. However, there tended to be a consistent pattern across outcomes relating to the direction of effect and effect size, which may suggest that the non-significant findings were derived from underpowered studies with larger confidence intervals, while there was greater precision in the estimate where statistically significant results were found. 

Maternal outcomes where there is potential for intervention benefit included excess and adequate GWG, postnatal weight retention, caesarean delivery, preeclampsia, hypertension and GDM, and infant outcomes included birthweight and preterm delivery. However, there was also a consistent pattern in the reported meta-analyses of no, or very few, statistically significant results of the effect of interventions on some infant health outcomes, which did not appear to be related to sample size. These were low or high birthweight, NICU admission, low Apgar score, and mortality outcomes. As some of these outcomes are relatively rare, it may be that even the larger studies lacked the statistical power to detect a difference between the intervention and the control groups. Comparing the behaviours and population subgroups, the evidence base for smoking interventions suggests particular effectiveness for increasing birthweight, whereas diet-only interventions appear to be most effective at reducing GWG and physical-activity-only interventions were most effective for a reduction in GDM. When diet and physical activity systematic reviews reported a meta-analysis stratified by maternal BMI category, there was little impact on the pooled effect size for most maternal and infant outcomes, with the exception of GWG, where the largest reductions were observed among women with a BMI in the overweight or obese categories. However, most of the evidence-base identified in this review was related to diet and physical activity interventions, with only two systematic reviews providing meta-analysis evidence for smoking interventions and health outcomes, and no reviews reporting a meta-analysis of health outcomes and alcohol interventions. There was also a difference between intervention behaviours and the focus on specific outcomes reported, with diet and physical activity systematic reviews reporting a meta-analysis of both maternal and infant outcomes, whereas the smoking reviews only reported infant outcomes.

One explanation for some of the conflicting findings in the meta-analyses might be related to unmeasured factors. Implementing a behaviour change intervention to improve health-related outcomes requires women to change their behaviours first. By focusing on the health-related outcomes, we are missing essential behavioural information that has a direct impact on effectiveness. We discussed, in our previous paper [23], how important it is to measure behaviours to advance our understanding of the mechanisms within intervention research. If a behaviour-change intervention is not effective at reducing health-related risk, and the behaviour itself is not measured, then we are left questioning whether the behaviours are not relevant and we should cease trying to intervene to change that target behaviour. However, the lack of effect could be due to the intervention failing to change the target behaviour (either at all or to the magnitude required to have a clinically important impact on health outcomes), in which case, are there alternative behaviour-change strategies which could be explored with a greater potential to impact health outcomes? We previously reported [23] that systematic reviews reported the effectiveness of interventions at changing maternal behaviours in 100% of the smoking and alcohol reviews identified, but for only 18% of diet and/or physical activity reviews. We see the opposite trend in this paper, where 66% of diet and/or physical activity systematic reviews reported a meta-analysis of health-related outcomes, compared with only 9% of the smoking reviews and no alcohol reviews. This mismatch in priorities between researchers regarding different behaviours requires further attention, as exploring both maternal behaviours and health-related outcomes are important for smoking, alcohol, diet and physical activity interventions in pregnancy. We must also consider the influence of intervention components on the pooled data; can we identify which components appear to have the strongest influence on maternal behaviour and health outcomes? For example, how important is the intervention’s timing, intensity, and delivery mechanism? Are there commonalities across behaviours? We identified 120 unique citations for included interventions in the two smoking systematic reviews, and 675 in the 63 diet and physical activity reviews. Given the high volume of interventions published to date, and as more protocols are appearing for new interventions, which do not appear to differ substantially from those which already exist, we need to reflect on how best to drive this field forward. We have a unique window of opportunity offered by pregnancy to improve short- and long-term outcomes for mothers and infants, and to capitalise on this we must better understand how interventions do, or do not, work, rather than repeat what others have previously done. This type of repetition amounts to research waste and offers little to the scientific community. The third paper in this wider programme of research will explore factors such as intervention components and modes of delivery, across behaviours, to identify whether there is an existing evidence-base looking at how these impact the effectiveness of interventions, and to identify gaps for future research directives.

When looking at the consistency in the evidence-base across behaviours, we identified a high volume of diet and physical activity systematic reviews (and included intervention studies within the reviews) that report maternal- and infant-health-related outcomes, whereas there was a lack of evidence for alcohol interventions and limited evidence for smoking cessation. When comparing the diet and physical activity and smoking evidence-base that we did identify, there were some consistencies relating to elements of the methods employed, and to the context of the evidence-base included in the reviews. There was a pattern of systematic reviews having strong search strategies with additional searches to supplement their database searches, and a consistency in the range of quality scores and reporting of conflicts of interest. There were similar publication date ranges for all types of behaviours, and some consistency in the intervention settings in the High-Income and Upper-Middle-Income Countries. However, there was a lack of any smoking interventions set in Lower-Middle-Income or Low-Income Countries, whereas there was some, albeit limited, representation of diet and physical activity interventions. 

This systematic review of systematic reviews employed rigorous methods. We conducted extensive searches of bibliographic databases and additional data sources. All screening was carried out in duplicate, and data extraction and quality assessments were validated using standardised protocols. The protocol for the review was published as a peer-reviewed paper [26] and on PROSPERO (CRD42016046302). There were some deviations from the published protocol for this paper, primarily relating to the exclusion of systematic reviews that did not report a meta-analysis of at least two studies, whereas our previous published paper on maternal behaviour outcomes also included narrative syntheses [23]. This decision was made based on the focus of this paper being health-related outcomes, which usually have a high degree of standardisation in reporting, making them suitable for pooling in meta-analysis. Conversely, our previous paper [23] focused on maternal behaviours as outcomes (e.g., energy intake, fruit and vegetable consumption, smoking cessation), which have a much less standardised method of estimating, and the need for standardisation across studies to facilitate meta-analysis was a recommendation we made [23].

As with all systematic reviews, we were limited by the availability and quality of data. This meant that we were not able to explore the effectiveness of alcohol interventions on health-related outcomes at all, and we were limited to just two reviews reporting infant health outcomes for smoking. One of the aims of a systematic review of systematic reviews is to describe the current extent of the evidence and the gaps in it to inform future research. We have identified clear gaps in the current evidence base which warrant further research. The quality of the included systematic reviews was also considered to be good, with no reviews being categorised as low-quality overall. However, with only two smoking systematic reviews, the summary percentage scores for individual questions can be heavily influenced by a single review and should be interpreted with caution. There was a lack of data reported by the included systematic reviews describing the ethnicity of the participants or exploring the effectiveness of interventions according to ethnic groups, which is a limitation, especially given the diversity of countries represented. Additionally, most intervention data originated from High-Income Countries, followed by Upper-Middle-Income Countries, which is a major limitation. The results of this systematic review of systematic reviews are not likely to be relevant to Low- or Lower-Middle-Income countries for smoking interventions, and there is likely to be limited relevance regarding diet and physical activity interventions due to their minimal representation. This is an important limitation of the existing evidence-base and there is a pressing need for more research in Low- and Lower-Middle-Income Countries. Non-communicable diseases account for approximately 70% of deaths globally, with almost double the rate among adults in Low- and Lower-Middle-Income Countries compared with adults in High-Income Countries [94], and diet and tobacco behaviours are among the top three risk factors for global causes of death [95]. 

This systematic review of systematic reviews has identified the extent of current research across four behaviours and, importantly, the gaps in the evidence base which can inform future research activities. The most consistent data relate to intervention impact on maternal-weight-related outcomes, with some promising evidence from the larger meta-analyses for additional outcomes, such as caesarean delivery and GDM. We have identified evidence gaps for meta-analysis of alcohol interventions and health-related outcomes, limited evidence for smoking interventions, and for interventions set in Lower-Middle-Income and Low-Income Countries which require further research. Where a high volume of evidence already exists, in this case, diet and physical activity interventions, there needs to be a shift in research focus to advance this field. This could include reporting both maternal behaviours and health outcomes simultaneously, and exploring intervention components which influence the effectiveness of interventions in order to advance our understanding of the mechanisms, and enable researchers and practitioners to capitalise on the unique opportunity that pregnancy presents for short- and long-term gains in maternal and infant health. 

## Figures and Tables

**Figure 1 nutrients-13-01036-f001:**
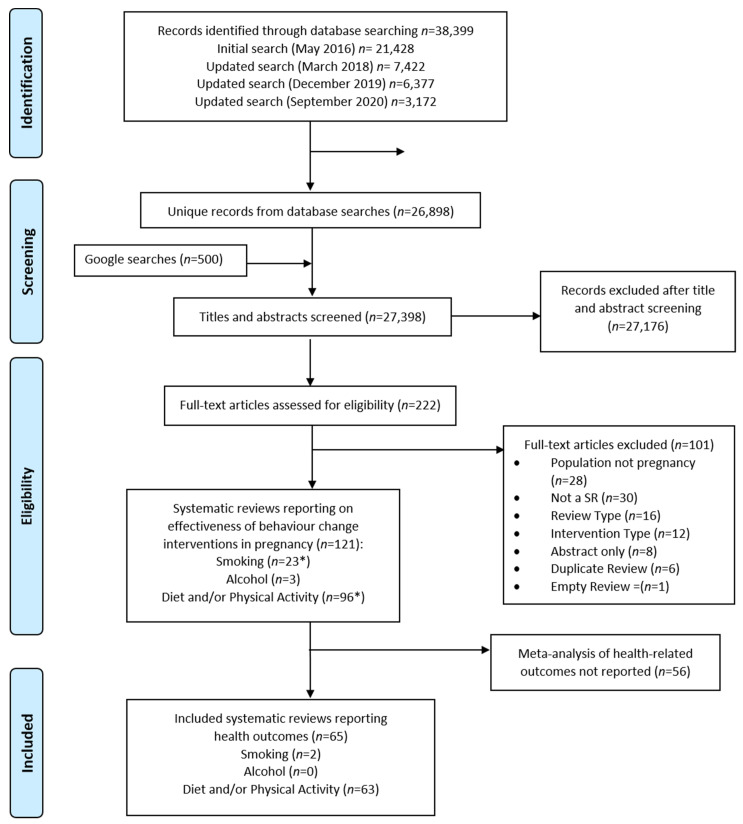
PRISMA Flowchart. * One systematic review included both smoking and diet and/or physical activity intervention studies.

**Figure 2 nutrients-13-01036-f002:**
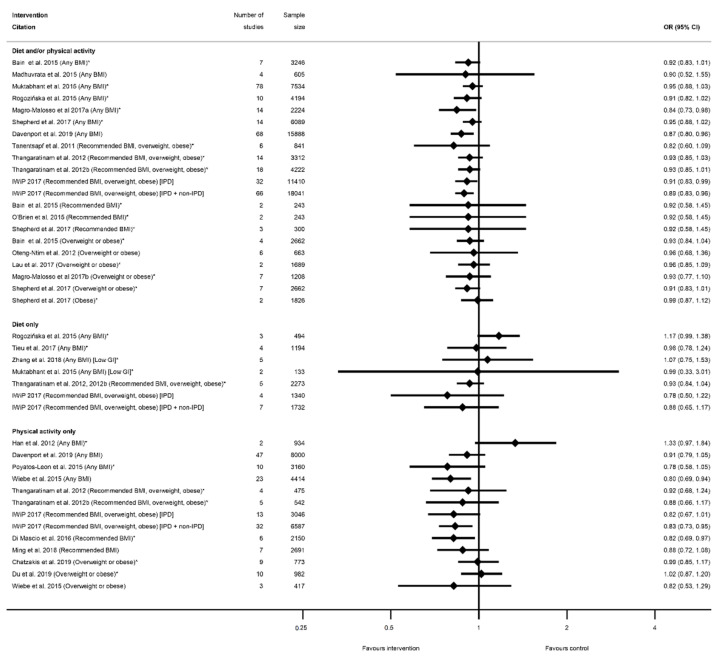
Forest plot of meta-analysis results for total gestational weight gain (GWG). * indicates the estimate is relative risk.

**Figure 3 nutrients-13-01036-f003:**
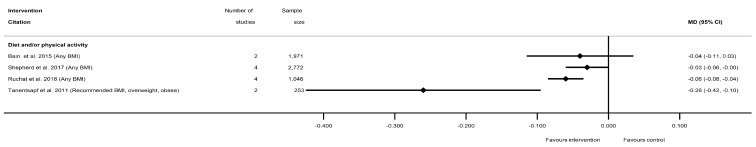
Forest plot of meta-analysis results for weekly gestational weight gain (GWG).

**Figure 4 nutrients-13-01036-f004:**
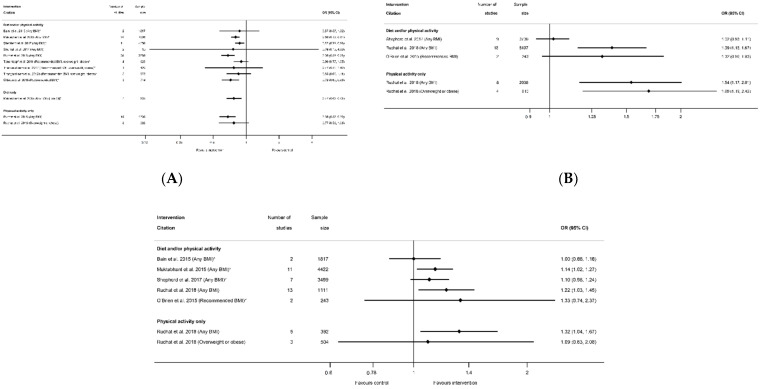
Forest plot of meta-analysis results for excess, adequate and inadequate gestational weight gain (GWG); (**A**) excessive GWG, (**B**) adequate GWG, (**C**) inadequate GWG. * indicates the estimate is relative risk.

**Figure 5 nutrients-13-01036-f005:**
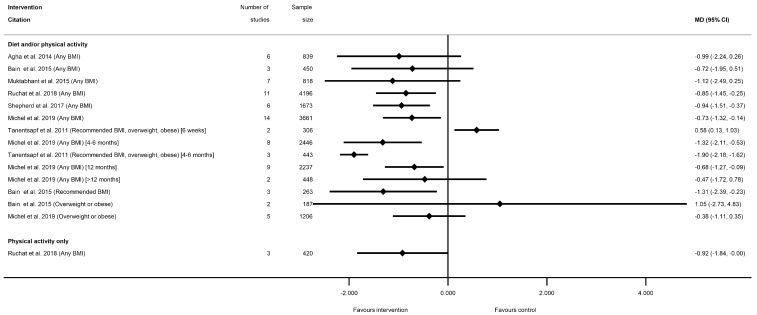
Forest plot of meta-analysis results for postnatal weight retention.

**Figure 6 nutrients-13-01036-f006:**
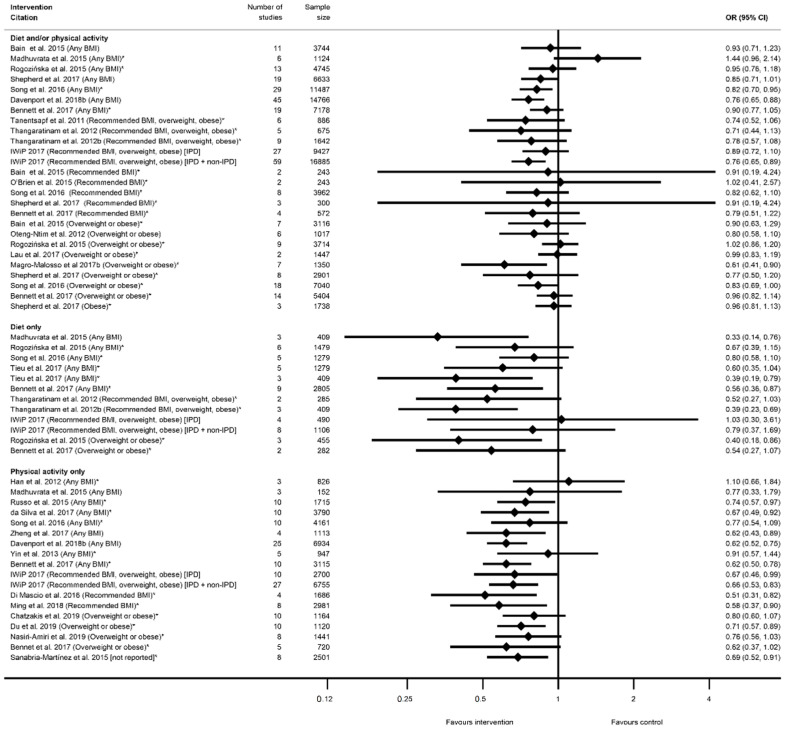
Forest plot of meta-analysis results relating to gestational diabetes (GDM). * indicates the estimate is relative risk.

**Figure 7 nutrients-13-01036-f007:**
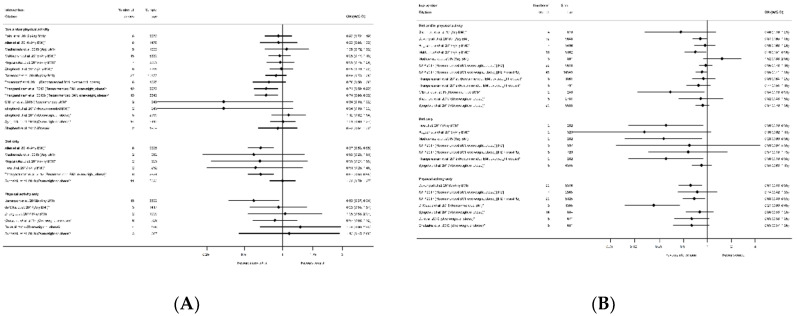
Forest plot of meta-analysis results relating to hypertensive disorders; (**A**) preeclampsia, (**B**) pregnancy induced hypertension. * indicates the estimate is relative risk.

**Figure 8 nutrients-13-01036-f008:**
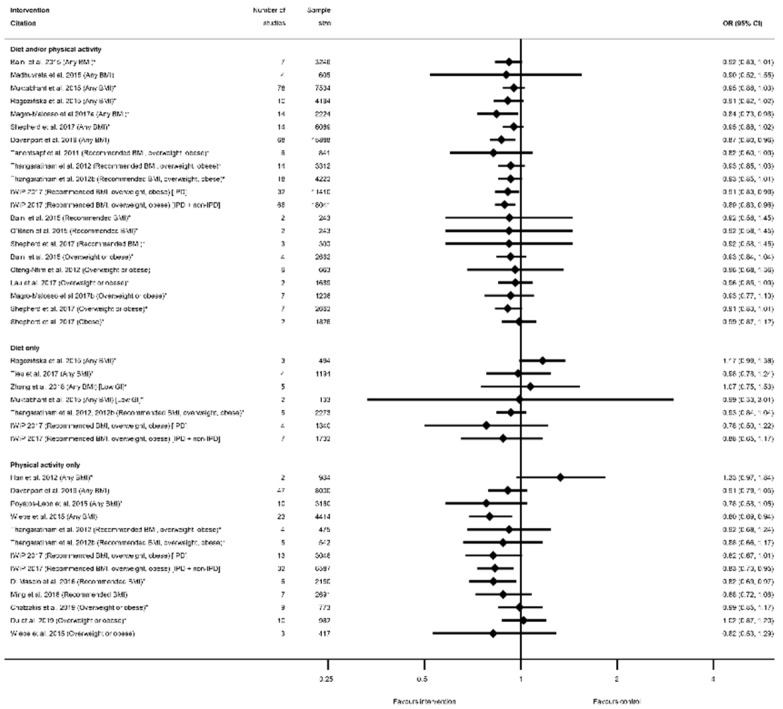
Forest plot of meta-analysis results for caesarean delivery. * indicates the estimate is relative risk.

**Figure 9 nutrients-13-01036-f009:**
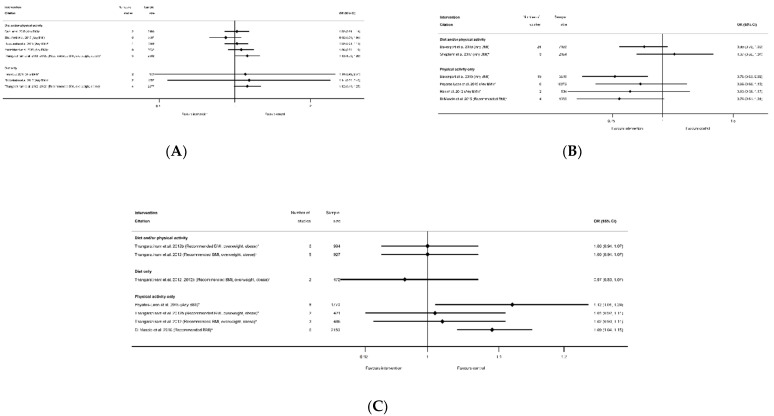
Forest plot of meta-analysis results for other modes of delivery; (**A**) Induction of labour, (**B**) instrumental vaginal delivery and (**C**) vaginal delivery. * indicates the estimate is relative risk.

**Figure 10 nutrients-13-01036-f010:**
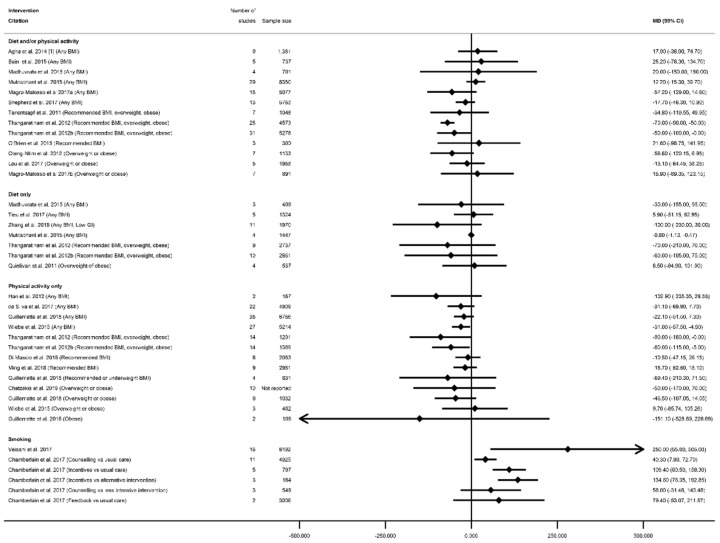
Forest plot of meta-analysis results for birthweight.

**Figure 11 nutrients-13-01036-f011:**
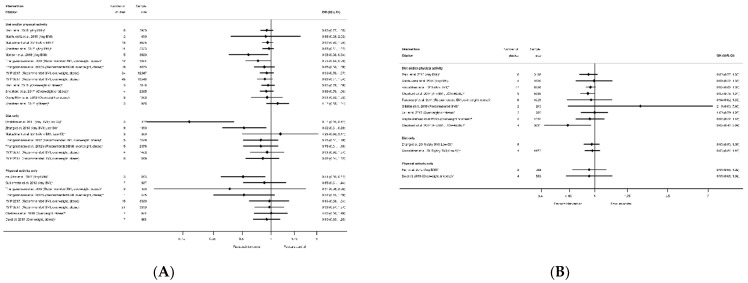
Forest plot of meta-analysis results for high birthweight outcomes; (**A**) LGA, (**B**) macrosomia. * indicates the estimate is relative risk.

**Figure 12 nutrients-13-01036-f012:**
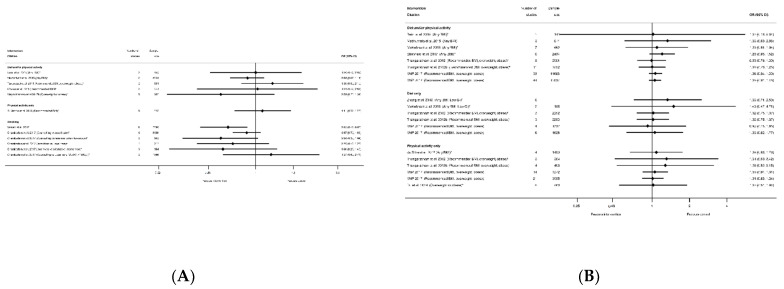
Forest plot of meta-analysis results for low-birthweight outcomes (**A**) LBW, (**B**) SGA. * indicates the estimate is relative risk.

**Figure 13 nutrients-13-01036-f013:**
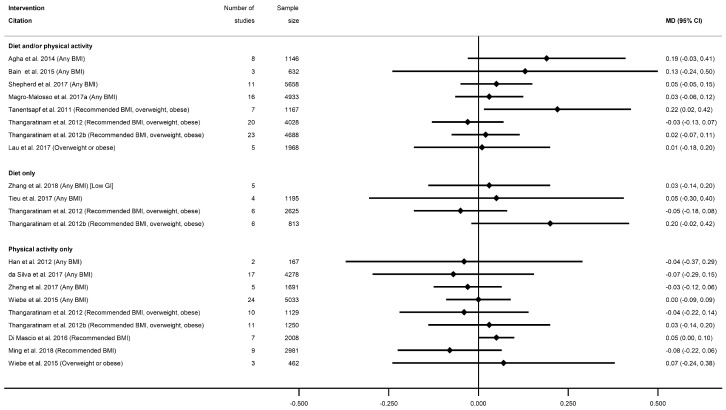
Forest plot of meta-analysis results for gestational age at delivery.

**Figure 14 nutrients-13-01036-f014:**
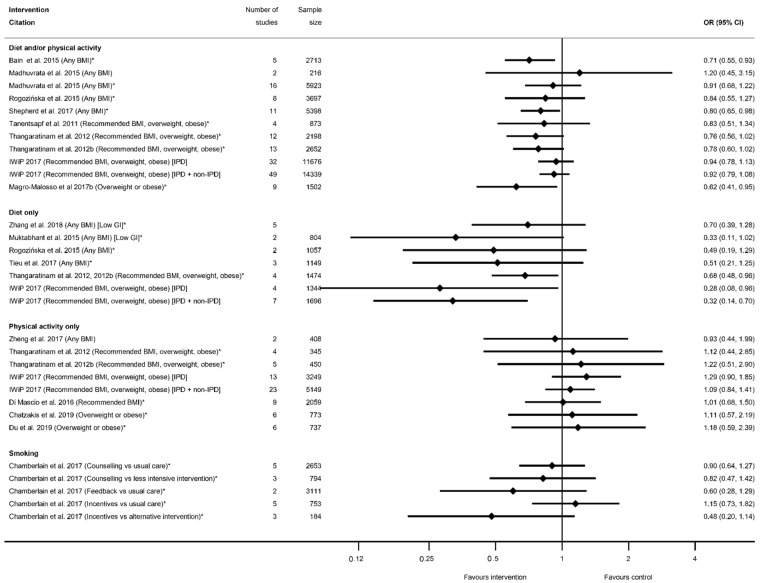
Forest plot of meta-analysis results for preterm delivery. * indicates the estimate is relative risk.

**Figure 15 nutrients-13-01036-f015:**
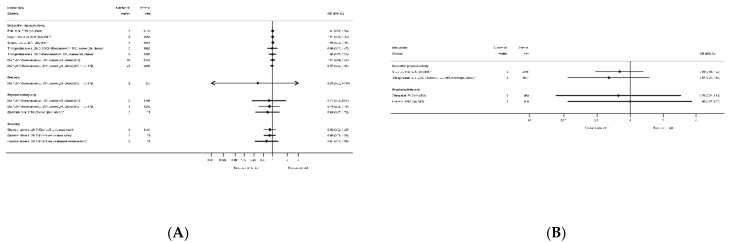
Forest plot of meta-analysis results for (**A**) Neonatal Intensive Care Unit (NICU) admission and (**B**) low Apgar score. * indicates the estimate is relative risk.

**Table 1 nutrients-13-01036-t001:** Summary data for included systematic reviews.

Systematic Reviews Reporting Smoking Interventions (*n* = 2)		Systematic Reviews Reporting Diet and/or Physical Activity Interventions (*n* = 37)	Systematic Reviews Reporting Diet-only Interventions (*n* = 19)	Systematic Reviews Reporting Physical-Activity-only Interventions (*n* = 37)
Time period of included intervention studies in the reviews				
**Range in years:**	1976 to 2016	1974 to 2018	1975 to 2018	1974 to 2019
Search strategies:				
**Databases only (*n*)**	0	3	1	2
**Supplementary searches (*n*)**	2	34	18	35
Included study designs:				
**RCT only (*n*)**	2	24	18	27
**RCT + other design (*n*)**	0	13	1	10
Number included studies in the systematic reviews				
**Range (*n*)**	16 to 88	5 to 113	4 to 89	3 to 113
**Median (*n*)**	N/A	21 (IQR 14 to 46)	23 (IQR 14 to 44)	23 (IQR 10 to 49)
Pooled sample sizes included in the systematic reviews				
**Range (*n* women)**	6192 to >26,000	363 to 273,182	537 to 182,139	214 to 598,185
**Median (*n* women)**	N/A	6920 (IQR 2748 to 12,007)	8558 (IQR 2786 to 11,487)	4350 (IQR 1768 to 14,601)
**Not reported (*n*)**	1 *	0	0	1 *
**Countries of intervention studies included in the systematic reviews (reported for *n* systematic reviews)**		Not reported for 6 systematic reviews	Not reported for 6 systematic reviews	Not reported for 7 systematic reviews
**HICs represented in the included interventions**	1. Australia 2. Canada 3. Denmark 4. France 5. Greece 6. Ireland 7. Netherlands 8. New Zealand 9. Norway 10. Poland 11. Spain 12. Sweden 13. UK 14. USA	1. Australia 2. Belgium 3. Canada 4. Croatia 5. Denmark 6. Finland 7. France 8. Germany 9. Greece 10. Ireland 11. Italy 12. Japan 13. Netherlands 14. New Zealand 15. Norway 16. Poland 17. Portugal 18. South Korea 19. Spain 20. Sweden 21. Switzerland 22. Taiwan 23. UK 24. USA	1. Australia 2. Belgium 3. Canada 4. Chile 5. Denmark 6. Finland 7. Germany 8. Greece 9. Ireland 10. Italy 11. Netherlands 12. New Zealand 13. Norway 14. Spain 15. Sweden 16. Taiwan 17. UK 18. USA	1. Australia 2. Belgium 3. Canada 4. Croatia 5. Denmark 6. Finland 7. France 8. Germany 9. Greece 10. Ireland 11. Italy 12. Japan 13. Netherlands 14. New Zealand 15. Norway 16. Poland 17. Portugal 18. South Korea 19. Spain 20. Sweden 21. Switzerland 22. Taiwan 23. UK 24. USA
**UMICs represented in the included interventions**	1. Argentina 2. Brazil 3. Columbia 4. Cuba 5. Mexico	1. Argentina 2. Brazil 3. China 4. Colombia 5. Iran 6. Kosovo 7. Mexico 8. Serbia 9. South Africa 10. Thailand 11. Turkey	1. Argentina 2. Brazil 3. China 4. Columbia 5. Indonesia (East Java) 6. Iran 7. Kosovo 8. Mexico 9. South Africa 10. Turkey	1. Argentina 2. Brazil 3. China 4. Colombia 5. Iran 6. Kosovo 7. Mexico 8. Serbia 9. South Africa 10. Thailand 11. Turkey
**LMICs represented in the included interventions**	None	1. Benin 2. Egypt 3. India 4. Pakistan	1. Egypt 2. India	1. Benin 2. Egypt 3. India 4. Pakistan
**LICs represented in the included interventions**	None	None	1. Gambia	None

Footnote: Income status of the countries defined according to the World Bank data for the current 2020 fiscal year “low-income economies are defined as those with a GNI per capita, calculated using the World Bank Atlas method, of $1025 or less in 2018; lower middle-income economies are those with a GNI per capita between $1026 and $3995; upper middle-income economies are those with a GNI per capita between $3996 and $12,375; high-income economies are those with a GNI per capita of $12,376 or more” [93]. Abbreviations: HICs = High Income Countries; UMICs = Upper Middle-Income Countries; LMICs -= Lower Middle-Income Countries; LICs = Lower Income Countries; IQR = Interquartile Range; N/A = not applicable to calculate median with only two systematic reviews.* One smoking systematic review reported >26,000 women but not the exact number; one physical activity only systematic review reported the number of neonates rather than the number of women.

**Table 2 nutrients-13-01036-t002:** Summary of quality assessment of included systematic reviews.

Intervention Type	Quality Assessment Question											Total Score	Quality Category
	1	2	3	4	5	6	7	8	9	10	11		
Smoking interventions	100% (2/2)	100% (2/2)	100% (2/2)	100% (2/2)	50% (1/2)	0% (0/2)	100% (2/2)	100% (2/2)	100% (2/2)	50% (1/2)	50% (1/2)	Range 7–10	50% moderate (1/2) 50% high (1/2)
Diet and physical activity interventions	100% (37/37)	84% (31/37)	92% (34/37)	76% (28/37)	95% (35/37)	76% (28/37)	86% (32/37)	100% (37/37)	71% (26/37)	86% (32/37)	84% (31/37)	Range 6–11	8% moderate (3/37) 92% high (34/37)
Diet-only interventions	100% (19/19)	74% (14/19)	89% (17/19)	79% (15/19)	89% (17/19)	79% (15/19)	84% (16/19)	100% (19/19)	89% (17/19)	89% (17/19)	89% (17/19)	Range 6–11	16% moderate (3/19) 84% high (16/19)
Physical-activity-only interventions	100% (37/37)	81% (30/37)	95% (35/37)	84% (31/37)	89% (33/37)	59% (22/37)	81% (30/37)	100% (37/37)	70% (26/37)	78% (29/37)	78% (29/37)	Range 6–11	14% moderate (5/37) 86% high (32/37)
Total for all intervention types *	100% (65/65)	86% (56/65)	94% (61/65)	85% (55/65)	89% (58/65)	69% (45/65)	85% (55/65)	100% (65/65)	72% (47/65)	85% (55/65)	78% (51/65)	Range 6–11	12% moderate (8/65) 88% high (57/65)

Footnote: quality assessment questions were: (1) Is the review question clearly and explicitly stated?; (2) Were the inclusion criteria appropriate for the review question?; (3) Was the search strategy appropriate?; (4) Were the sources and resources used to search for studies adequate?; (5) Were the criteria for appraising studies appropriate?; (6) Was critical appraisal conducted by two or more reviewers independently?; (7) Were there methods to minimize errors in data extraction?; (8) Were the methods used to combine studies appropriate?; (9) Was the likelihood of publication bias assessed?; (10) Were recommendations for policy and/or practice supported by the reported data?; (11) Were the specific directives for new research appropriate?. For total score 1 is given if yes otherwise it is zero. For quality: Low quality is 0–3. Moderate quality is 4–7. High quality is 8–11. * Note—some of the same systematic reviews were included in diet and or/physical activity, diet-only and physical-activity-only summaries depending on whether they reported data for all or a combination of intervention types, whereas the total quality summary for all included systematic reviews only includes each systematic review once (out of a total of 59 included reviews).

**Table 3 nutrients-13-01036-t003:** Summary of meta-analysis of maternal health outcomes reported by the included systematic reviews.

Categories of Maternal Outcomes	Systematic Reviews Reporting Meta-Analysis	Number of Meta-Analyses Reported	Smoking Interventions (Outcomes Reported; Number of Meta-Analyses)	Diet and/or Physical Activity Interventions (Outcomes Reported; Number of Meta-Analyses)	Diet-Only Interventions (Outcomes Reported; Number of Meta-Analyses)	Physical-Activity-Only Interventions (Outcomes Reported; Number of Meta-Analyses)
**Maternal weight-related outcomes (Appendix A)**	38	114	None reported	Total GWG; 32 Weekly GWG; 4 Excess GWG; 9 Inadequate GWG; 5 Adequate GWG; 3 Postnatal weight retention; 15 Other related outcomes; 4	Total GWG; 10Excess GWG; 1	Total GWG; 24Excess GWG; 2Inadequate GWG; 2Adequate GWG; 2Postnatal weight retention; 1
**GDM-related outcomes (Appendix A)**	32	73	None reported	GDM diagnosis; 2 6Other related outcomes; 2	GDM diagnosis; 14 Other related outcomes; 6	GDM diagnosis; 19 Other related outcomes; 6
**Hypertensive disorders of pregnancy-related outcomes (** Appendix A **)**	22	59	None reported	Pre-eclampsia; 15 Hypertension; 12 Other related outcomes; 3	Pre-eclampsia; 6 Hypertension; 8 Other related outcomes; 2	Pre-eclampsia; 6 Hypertension; 7
**Mode of delivery-related outcomes (Appendix A)**	25	63	None reported	Caesarean delivery; 21 Induction of labour; 5 Instrumental delivery; 2 Vaginal delivery; 2	Caesarean delivery; 8 Induction of labour; 3 Vaginal delivery; 1	Caesarean delivery; 13 Instrumental delivery; 4 Vaginal delivery; 4
**Other maternal outcomes (Appendix A)**	8	23	None reported	PPH; 3Composite outcome; 1 Low back pain; 1 Perineal trauma; 1 Prenatal mental health; 5 Postnatal mental health; 3	PPH; 1 Composite outcome; 1	Composite outcome; 1 Prenatal mental health; 4 Postnatal mental health; 2
**Total**	58 systematic reviews *	334 meta-analyses	0 meta-analyses	176 meta-analyses	61 meta-analyses	97 meta-analyses

* Total number of unique systematic reviews that reported maternal outcomes, where some reviews reported multiple meta-analyses for multiple maternal outcomes. GDM—gestational diabetes mellitus, GWG—gestational weight gain, PPH—postpartum haemorrhage.

**Table 4 nutrients-13-01036-t004:** Summary of meta-analysis of maternal health outcomes reported by the included systematic reviews.

Categories of Infant Outcomes	Systematic Reviews Reporting Meta-Analysis	Number of Meta-Analyses Reported	Smoking Interventions (Outcomes Reported; Number of Meta-Analyses)	Diet and/or Physical Activity Interventions (Outcomes Reported; Number of Meta-Analyses)	Diet-Only Interventions (Outcomes Reported; Number of Meta-Analyses)	Physical-Activity-Only Interventions (Outcomes Reported; Number of Meta-Analyses)
**Fetal growth-related outcomes (** Appendix A **)**	33	150	Birth weight; 6 Low birth weight; 6	Birth weight; 14 LGA; 13Macrosomia; 9 SGA; 8 Low birth weight; 5 Other fetal growth; 10	Birth weight; 9 LGA; 8Macrosomia; 4 SGA; 7 Low birth weight; 1 Other fetal growth; 10	Birth weight; 16 LGA; 8Macrosomia; 3 SGA; 6 Low birth weight; 1 Other fetal growth; 6
**Gestational age at delivery-related outcomes (** Appendix A **)**	26	55	Preterm; 5	Gestational age; 8 Preterm; 11	Gestational age; 5 Preterm; 8	Gestational age; 10 Preterm; 8
**Mortality-related outcomes (** Appendix A **)**	10	17	Stillbirth; 2 Neonatal mortality; 1	Stillbirth; 4 Intrauterine death; 1 Neonatal mortality; 1 Perinatal mortality; 2 Miscarriage; 1	Stillbirth; 1 Perinatal mortality; 1	Intrauterine death; 1 Perinatal mortality; 1 Miscarriage; 1
**NICU (** Appendix A **)**	7	14	Admission; 3	Admission; 7	Admission; 1	Admission; 3
**Apgar score (** Appendix A **)**	7	11	None reported	<7 at 5 minutes; 2 Score at 1 minute; 1 Score at 5 minutes; 2	<7 at 5 minutes; 1 Score at 5 minutes; 1	<7 at 5 minutes; 2 Score at 1 minute; 1 Score at 5 minutes; 1
**Other infant health-related outcomes (** Appendix A **)**	8	23	None reported	Shoulder dystocia; 4 Hypoglycaemia; 3 Respiratory distress; 3 Infant hyperbilirubinemia; 2 Birth trauma; 2 PROM; 1 Breastfeeding; 2 Composite outcome; 1	Shoulder dystocia; 1 Hypoglycaemia; 1 Placental weight; 1 Composite outcome; 1	Composite outcome; 1
**Total**	36 *	270	23	117	61	69

* Total number of unique systematic reviews that reported maternal outcomes, where some reviews reported multiple meta-analyses for multiple maternal outcomes. LGA—large for gestational age, SGA—small for gestational age, NICU—neonatal intensive care unit, PROM—premature rupture of membranes.

## Data Availability

All data are reported within the manuscript and supplementary materials.

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
