# Peer review of "The Effectiveness of Smoking Cessation, Alcohol Reduction, Diet and Physical Activity Interventions in Improving Maternal and Infant Health Outcomes: A Systematic Review of Meta-Analyses"

_nutrients, 2021, doi:10.3390/nu13031036_

Round 1

Reviewer 1 Report

This is a systematic review of systematic reviews of antenatal behavioural interventions (targeted at alcohol consumption, smoking cessation, diet and physical activity) aiming to improve maternal and infant outcomes. The overall methodology is sound and adequately reported. I do have some major concerns about the reporting of results and some more minor suggestions/comments.

The most substantial concern is the overemphasis on statistical significance, and an apparent misunderstanding of what statistical significance means, and how it relates to sample size. This is reflected in (a) the dichotomizing of results into 'significant' and 'not significant', (b) the reporting and discussion only of statistically significant effects, with non-significant effect estimates ignored; (c) the repeated observation that the statistically significant studies tended to be the ones with the larger sample sizes. I get the impression that the authors think that statistical significance means that an estimated effect is "real" (and that non-significant results are somehow less valid), and that statistical significance correlating with larger sample sizes means that the statistically significant results are more likely to be 'true' than the non-statistically significant ones.

Firstly, statistical significance is an arbitrary cut-point imposed on a continuous measure. I assume that the authors are using the traditional p<0.05 definition, though they nowhere state this. An estimate with a p value of 0.049 and an estimate with a p value of 0.051 in fact present pretty much the same level of evidence of an effect. There is no justification for treating them as different in kind and discussing only the first.

Secondly, if the methodology used to derive the effect estimates was sound, all of the estimates are valid estimates of the true effect size whether they get under the magic 0.05 threshold or not, and there is no justification for simply ignoring the non-significant ones in the discussion.

Thirdly, there is nothing at all surprising in the fact, repeatedly noted, that the statistically significant estimates tend to be from studies with larger sample sizes. Statistical significance is a function of sample size. What a larger sample size does is allow you to obtain a more precise estimate, i.e., the standard error of the estimate will be smaller. The true effect size for any intervention will pretty much never be exactly zero; but the closer to zero it is, the larger the sample size you need to differentiate it from zero. So the larger studies being statistically significant while the smaller ones aren't implies at most that the size of the effect isn't very large, such that the smaller studies are underpowered to detect it. Conversely, when you get statistically significant results from smaller samples and non-statistically-significant results from larger ones, it suggests that the true effect is very small, and that the statistically significant results are substantial over-estimates of the true effect size (they happened to get a sample in which the observed effect was much larger than the true population effect).

So it is worth discussing the role of sample size in explaining why some studies have smaller p values and narrower confidence intervals than other studies, but the relevance of sample size is in saying whether (a) statistically significant results are more likely to be flukes occurring in underpowered studies, and (b) whether statistical significance is actually interesting given that it only occurs in larger sample sizes (and then presumably is of fairly small magnitude).  It is much more interesting to focus instead on the consistency of the effect estimates: are the point estimates in the same direction; are they of a consistent magnitude, and is this magnitude one which is actually important?

The paper therefore needs to discuss the 'non-significant' results alongside the 'significant' ones, amend the discussion of the sample sizes vis-a-vis statistical significance, and add some discussion of whether the actual estimated effects (from both 'significant' and 'non-significant' results) are likely to be clinically meaningful.

Other comments:

  1. The abstract does not mention alcohol consumption, even though it is in the title and is one of the interventions considered. Even though you didn't find any relevant SRs, it should be mentioned here.

  2. Why choose alcohol consumption, smoking, diet and physical activity as the relevant behaviours? (As opposed to, e.g. illicit drug use, avoiding fish/soft cheese, folate +iodine supplementation or other behaviours that are known to relate to adverse pregnancy outcomes)?

  3. Lines 45-47 twice mention obesity. This creates the impression that the authors think that (a) obesity is a behaviour, and (b) that diet and physical activity interventions are only needed for women with obesity (and maybe that GDM only occurs in obese women?). Obviously obesity is not a behaviour and while 'influenced by' diet and physical activity it is not equivalent to (poor) diet and physical activity behaviours. Conversely, diet and physical activity interventions are relevant to women of all BMIs, and women of all BMIs can experience adverse pregnancy and birth outcomes (including GDM) related to poor diet and physical activity. Moreover the SRs of diet and physical activity interventions are not just focused on women with obesity. Perhaps the authors mean to point out that risks of some adverse outcomes are higher in women with obesity, and that interventions might therefore be especially relevant in this sub-population.

  4. Lines 54-55, and also elsewhere throughout the paper: the reference to "behavioural domains" is confusing. I gather that the authors mean the different behaviours, i.e., smoking, alcohol consumption, diet, physical activity. But then what are "similarities between behavioural domains in the prevention of adverse health outcomes"?

  5. Lines 105-106: "Systematic reviews were not excluded based on the type of comparator groups." What does this mean?

  6. Line 141: "low glycaemic index" presumably means "low glycaemic index diet".

  7. PRISMA flowchart: in the box 'Full-text articles excluded', one reason is 'Participants' - what does this mean? (Not an antenatal population?)

  8. Table 1: The first column is not labelled; I can see that it refers to the SRs of smoking cessation, but it still should have a column header. Also in this table:

    1. Included study designs has a row saying "RCT + other design" - what is the other design and how is it added to the RCT?

    2. In 'pooled sample sizes included', you say that the median (n women) is not possible to calculate; better to say median is not really meaningful for only 2 studies. (Also on line 200, the 'median' of 56 is reported for the number of studies in the smoking SRs; this should be omitted as it's not sensible to calculate a median for 2 values).

  9. The categorisation of studies primarily investigating GWG as related to 'maternal weight' is a bit misleading. GWG is not really a measure of 'maternal weight', as it includes, among other things, the weight of the fetus, increased blood and fluid volume, placental and breast tissue etc.

  10. LInes 258-259: sentence "There were no data available for any of the maternal health outcomes and meta-analysis of smoking interventions" is very confusing; from reading on I see that the smoking SRs did not include meta-analysis but am still not sure what the first part of the sentence means, since you report data for maternal health outcomes (e.g. pre-eclampsia).

  11. Table 3 (and elsewhere in discussion of results): it's not clear why you consider 'vaginal delivery' to be a different outcome from 'caesarean delivery': If caesarean delivery is No, then vaginal delivery is Yes, and vice versa. (Also not clear how 'instrumental delivery' fits, since it implies vaginal delivery but the 'no' category will include both NVD and C Section).

  12. Line 288 (and elsewhere, including in forest plots) refers to "recommended BMI". I assume you mean the WHO 'normal BMI' category, i.e. BMI 18.5-24.9 kg/m2,, but referring to it as 'recommended' is incorrect.

  13. References to 'excessive', 'adequate' and 'inadequate' GWG should note that these categories are as defined by the IoM (i.e. they are not some independent clinical reality; in fact the evidence base to support these categories is not as robust as often assumed).

  14. Axes on the forest plots need to be checked and fixed so that they represent sensible values. Firstly, the resolution of the graphs is too low and cannot be magnified out sufficiently to properly read the values on the axes; decimal points to the left of the null value in particular are very hard to see. Secondly, the points marked on the axes are often not symmetric around the null value (0 for MD, 1 for RR/OR)

  15. Line 361: as with C Section / Vaginal Delivery, it's not clear to me how 'postnatal weight loss' is actually a different outcome from 'postnatal weight retention'; presumably both are just a measure of postnatal weight compared to earlier (prenatal / pregnancy) weight?

  16. Line 421-422: RR estimates are reported without confidence intervals.

  17. Line 542: "macrosomia <400g" should be ">4000g".

  18. As with the points about statistical significance/ sample size above: the fact that statistical significance did not map onto larger studies for LGA, and that there were no statistically significant differences for mortality, NICU etc, is not surprising. These outcomes are relatively rare, and for binary outcomes the effective sample size is dependent on the number of events. A very large total sample is needed to get enough events to provide sufficient power to detect even moderately sized effects. If you happen to get a statistically significant effect with a smaller sample size it will usually be an overestimate of the true effect.

  19. Line 738-739: "there was also a consistent pattern in the reported meta-analyses for no effect of interventions on some infant health outcomes". A non-statistically-significant result is not equivalent to 'no effect'. It means that there is insufficient evidence for us to conclude (based on the predefined significance threshold) to conclude that there is an effect.

  20. I agree it's important to know if an intervention failed to work (where this is actually the result, rather than just lack of statistical significance!) because it failed to change behaviour, or because the behaviour change is not actually effective in changing outcomes. However, I'd disagree with the idea that it is only in the latter case that we should "cease trying to intervene." Even if a change in behaviour would affect outcomes, if interventions don't actually cause that change in behaviour, they are equally futile. The existence of some intervention that will cause the desired behaviour change cannot be assumed.

  21. Paragraph beginning on line 793: not clear to me how the discussion of SR methodology and its consistency or otherwise between diet/PA and alcohol/smoking interventions relates to "lack of evidence for alcohol interventions and limited evidence for smoking [cessation interventions]."

Reviewer 2 Report

A very useful review of an area of current importance. Overall the paper is well written and detailed methods described. I have a number of comments/questions.

There are some issues with the page numbering in th PDF.  

With respect to data extraction - could the authors confirm: 

Was the process blinded? 

How were any discrepencies resolved? 

I found table 1 interesting with repect to the range of countries the review covers, my understanding is that one review could cover a numebr of countries? Is this correct? 

There doesn't seem to be any ethnic differences mentioned - I would have found this interesting in relation to effectiveness of in particular dietary and PA interventions. 

Round 2

Reviewer 1 Report

n/a